# Transmembrane protein western blotting: Impact of sample preparation on detection of SLC11A2 (DMT1) and SLC40A1 (ferroportin)

**Yoshiaki Tsuji** [ID]*

Department of Biological Sciences, Toxicology Program, North Carolina State University, Raleigh, NC, United States of America

* ytsuji@ncsu.edu

## Abstract

Western blotting has been widely used for investigation of protein expression, posttranslational modifications, and interactions. Because western blotting usually involves heat-denaturation of samples prior to gel loading, clarification of detailed procedures for sample preparation have been omitted or neglected in many publications. We show here the case that even excellent primary antibodies failed to detect a specific protein of interest due to a routine heating practice of protein samples. We performed western blotting for transmembrane iron transporter proteins; SLC11A2 (divalent metal transporter 1, DMT1), SLC40A1 (ferroportin 1, Fpn1), and transferrin receptor-1 (TfR1), along with cytoplasmic iron storage protein ferritin H. Our results in 12 human culture cell lysates indicated that only unheated samples prior to gel loading gave rise to clear resolution of DMT1 protein, while heated samples (95˚C, 5min) caused the loss of resolution due to DMT1 protein aggregates. Unheated samples also resulted in better resolution for Fpn1 and TfR1 western blots. Conversely, only heated samples allowed to detect ferritin H, otherwise ferritin polymers failed to get into the gel. Neither different lysis/sample loading buffers nor sonication improved the resolution of DMT1 and Fpn1 western blots. Thus, heating samples most critically affected the outcome of western blotting, suggesting the similar cases for thousands of other transmembrane and heat-sensitive proteins.

## Introduction

More than 3,000 human genes encode transmembrane proteins that have been classified into receptors, transporters, enzymes, and other proteins [1]. Transmembrane proteins such as solute carrier (SLC) transporters play vital roles in import and export of nutrients, metabolites, and ions. Examples of SLC substrates are sugars, amino acids, vitamins, neurotransmitters, urea cycle metabolites, and metals [2, 3] including iron [4]. The human SLC transporter families comprise more than 400 genes and 52 subfamilies under the SLC family clusters, including MFS (major facilitator superfamily) and APC (amino acid-polyamine-organocation) superfamily [3, 5, 6]. The majority of SLC transporter proteins has 12-transmembrane segment (TMS) with diversities typically ranging from 9 to 14 TMS [7].

**Data Availability Statement:** All relevant data are within the paper and its Supporting Information files.

**Funding:** This work was supported in part by P30ES025128 from the National Institute of Environmental Health Sciences to the Center for

Human Health and the Environment (CHHE). The funder had no role in study design, data collection and analysis, decision to publish, or preparation of the manuscript.

**Competing interests:** The authors have declared that no competing interests exist.

Iron metabolism is coordinately regulated by iron import, export, and storage proteins localized in plasma membrane and intracellular compartments [8–10]. For instance, human transferrin receptor-1 (TfR1) is an 85 KDa homodimer of a plasma single-transmembrane protein serving as a major iron importing receptor for transferrin bound to two molecules of ferric iron ($Fe^{3+}$) [11, 12]. Iron is also subject to import and export by SLC transmembrane transporters. One is SLC11A1 and A2; particularly SLC11A2 (Divalent Metal Transporter—DMT1 or NRAMP2) which is a ubiquitously expressed transporter of iron along with cadmium, cobalt, manganese, and zinc to a lesser extent [13–15]. DMT1 plays a pivotal role in apical iron ($Fe^{2+}$) absorption (after $Fe^{3+}$ is reduced by ferric reductase DcytB, [16]) into duodenal enterocytes [13, 17, 18]. DMT1 also transports endosomal iron released from transferrin (and reduced by Steap3) into the cytoplasm for supply of ferrous iron ($Fe^{2+}$) in various cell types [8]. The DMT1 gene encodes four isoforms of mRNAs through alternative promoter and splicing [17, 19, 20], giving rise to at least four different DMT1 proteins [19]. The four isoforms of human DMT1 are 61–65 KDa transmembrane proteins having 12-TMS [13, 15, 21]. Another SLC protein in charge of iron export is SLC40A1 (Ferroportin 1—Fpn1 or IREG1) that is highly expressed in basolateral membrane of duodenal enterocytes and exports iron into blood stream [22–24], where $Fe^{2+}$ is oxidized to $Fe^{3+}$ by membrane-bound hephaestin or ceruloplasmin circulating in blood stream, transferring $Fe^{3+}$ to transferrin for systemic delivery of iron ($Fe^{3+}$) [25]. Fpn1 is also highly expressed in reticuloendothelial macrophages that release iron for recycling after erythrophagocytosis [26, 27]. Fpn1 is a 63 kDa multipass transmembrane protein predicted to contain 12-TMS [28, 29]. Hepcidin, a 25-amino acid peptide secreted from the liver in response to iron overload and inflammation [30–32], binds and induces internalization and degradation of Fpn1 leading to decrease in plasma iron levels and iron distribution to tissues [33]. When $Fe^{2+}$ levels are more than what cells need, the surplus of iron is stored into ferritin shell composed of 24 multi-subunits of tissue-specific ratios of H (heavy) and L (light) chains [11, 34, 35]. Iron ($Fe^{2+}$) is stored in ferritin shell as $Fe^{3+}$ catalyzed by the ferroxidase activity of the ferritin H subunit [34].

Western blotting is an important technique to investigate protein expression, posttranslational modifications (e.g. phosphorylation), and interactions with other proteins and macromolecules, in which we mostly use commercially available antibodies for detection of proteins of our interests. There are numerous monoclonal and polyclonal antibodies successfully used for western blotting and other applications (CiteAb, https://www.citeab.com/), while we still experience antibodies not working or far below our expectation of qualities for unknown reasons. To verify the quality and specificity of antibodies, we should take positive and negative control samples such as cell lysates prepared after transient transfection of expression plasmids or knockdown of endogenous mRNAs. In particular, detection of transmembrane proteins in western blotting has often been challenging due to difficulties in solubilization of those proteins with detergents and reducing agents. There are numerous numbers of transmembrane protein western blots so far published, including DMT1 and Fpn1 proteins (CiteAb). In this report, we initially judged our DMT1 and Fpn1 antibodies not working in our routine western blotting protocol; however, reassessment and modification of our protocol in a sample preparation step significantly improved the resolution of western blots.

## Materials and methods

### Cell culture and maintenance

12 human cell lines used in this study and their culture media are listed in Table 1. All cell lines were cultured at 37°C in a humidified $CO_2$ incubator (5% $CO_2$, Sanyo model MCO-17AIC). Undifferentiated Caco-2 cells were used in this study. Adherent cells were passaged by

**Table 1. All chemicals and reagents used in this study.**

| REAGENT OR RESOURCE | SOURCE | IDENTIFIER (catalog #) |
| --- | --- | --- |
| *Antibodies* | | |
| Mouse monoclonal anti-ferritin H | Santa Cruz Biotechnology | sc-135667 (3F8) |
| Rabbit polyclonal anti-transferrin receptor | Abcam | ab84036 |
| Rabbit monoclonal anti-DMT1/SLC11A2 | Cell Signaling Technology | 15083 (D3V8G) |
| Rabbit polyclonal anti-ferroportin 1(SLC40A1) | Novus Biologicals | NBP1-21502 |
| Mouse monoclonal anti-GAPDH | Chemicon | MAB374 |
| Mouse monoclonal anti-Flag | Sigma-Aldrich | F3165 |
| Goat anti-mouse IgG-peroxidase conjugated | Millipore | AP181P |
| Goat anti-rabbit IgG-peroxidase conjugated | Millipore | AP132P |
| *Chemicals/Culture Media* | | |
| Dulbecco's modified Eagle Medium (DMEM) | Corning | 50-003-PC |
| Minimum Essential Medium (MEM) | Corning | 50-011-PC |
| RPMI1640 | Corning | 50-020-PC |
| Ham's F12 | Corning | 50-040-PB |
| Opti-MEM | Life Technologies | 22600–134 |
| Penicillin Streptomycin solution, 100X | Corning | 30-002-CI |
| Sodium pyruvate | Corning | MT25000CI |
| Non-essential amino acid solution, 100X | Corning | 25-025-CI |
| Trypsin, 10X | Corning | 25-054-CI |
| Fetal Bovine Serum | Seradigm | 1400–500 |
| Ammonium iron(III) citrate (FAC) | Sigma-Aldrich | F5879 |
| Deferoxamine mesylate salt (DFO) | Sigma-Aldrich | D9533 |
| Hemin | Fluka | 51280 |
| Acrylamide | J.T. Baker | 4081–01 |
| Bis-Acrylamide | EMD Millipore | 2620 |
| Sodium dodecyl sulfate (SDS) | Calbiochem | 7910 |
| Sodium deoxycholate | Sigma-Aldrich | D-6750 |
| Nonidet-P40 (NP40) | US Biological | N-3500 |
| Tris (Hydroxymethyl) Aminomethane | J.T. Baker | 4109–06 |
| Glycine | J.T. Baker | 4057–06 |
| Sodium chloride | BDH | 7647-14-5 |
| Ammonium persulfate | Thermo Scientific | 17874 |
| TEMED | Fisher | BP150-100 |
| Skim milk powder | EMD Millipore | 1.15363.0500 |
| Bovine Serum Albumin (BSA) Fraction V | EMD Millipore (Calbiochem) | 2930 |
| Tween 20 | Fisher | BP337-500 |
| 2-mercaptoethanol | EMD Millipore | 6050 |
| Glycerol | Fisher | BP229-1 |
| Bromophenol blue | Sigma-Aldrich | B-8026 |
| Hepcidin | Peptide Institute, Inc. | PLP-4392-s |
| HyGLO Chemiluminescent HRP Detection Reagent | Denville Scientific | E2500 |
| Pierce ECL Western Blotting substrate | Thermo Scientific | 32106 |
| Clarity Western ECL Substrate | Bio-Rad | 170–5061 |
| Protein Assay Dye Reagent Concentrate | Bio-Rad | 5000006 |
| Precision Plus Protein Dual Color Standards | Bio-Rad | 161–0374 |
| Protease Inhibitor Cocktail Set I | Calbiochem/Millipore | 539131 |
| Transmembrane Protein Extraction Reagent | FIVEphoton Biochemicals | TmPER-50 |
| Nco I | New England Biolabs | R0193S |

*(Continued)*

**Table 1.** (Continued)

| REAGENT OR RESOURCE | SOURCE | IDENTIFIER (catalog #) |
|---|---|---|
| Hind III | New England Biolabs | R0104S |
| SmaI | New England Biolabs | R0141S |
| DNA polymerase Klnenow fragment | New England Biolabs | M0210L |
| Lipofectamine RNAiMax | Invitrogen | 13778–150 |
| *Cell lines* | *Source/Culture Media(containing 1x Penicillin Streptomycin)* | |
| HaCaT immortalized human keratinocyte | NE Fusenig/ DMEM high glucose (4.5g/L) +10%FBS | Boukamp et. al J. Cell Biol., 106: 761–771 (1988) |
| HepG2 human hepatocellular carcinoma | ATCC/ MEM +10% FBS | HB-8065 |
| HEK293 immortalized human embryonic kidney cells | ATCC/ MEM +10% FBS | CRL-1573 |
| HeLa human cervix adenocarcinoma | ATCC/ MEM +10% FBS | CCL-2 |
| K562 human erythroleukemia | ATCC/ RPMI1640, 25 mM HEPES +10%FBS | CCL-243 |
| SW480 human colon adenocarcinoma | ATCC/ DMEM high glucose (4.5g/L) +10%FBS | CCL-228 |
| A549 human lung carcinoma | Sigma-Aldrich/ DMEM high glucose +10%FBS | 86012804 |
| HL60 human promyelocytic leukemia | ATCC/ RPMI1640, 25 mM HEPES +10%FBS | CCL-240 |
| MCF7 human breast adenocarcinoma | ATCC/ MEM +10% FBS | HTB-22 |
| SH-SY5Y human neuroblastoma | ATCC/ MEM:Ham's F12(1:1) +10% FBS | CRL-2266 |
| Jurkat human acute T cell leukemia, clone E6-1 | ATCC/ RPMI1640, 25 mM HEPES, high glucose +10%FBS | TIB-152 |
| Caco-2 human colorectal carcinoma | ATCC/ MEM +20%FBS | HTB-37 |
| *plasmid DNA* | | |
| pCMVSPORT6 SLC40A1 | Dharmacon | MHS6278-202801506 |
| pBluescriptR SLC11A2(DMT1) | Open Biosystems | MHS1010-98075408 |
| pRK5SLC11A2(DMT1) | this work | |
| pCMVFlagSLC40A1(DMT1) | this work | |

trypsinization (0.25% Trypsin, 2.6 mM EDTA in calcium/magnesium free PBS) every 3–5 days when they were confluent. Suspension cells were passaged by 20~30-fold dilution to fresh growth media every 4–6 days.

## Chemicals, and reagents

All chemicals and reagents used in this study, their suppliers, and catalog numbers are listed in Table 1. Ferric ammonium citrate (FAC) and desferrioxamine (DFO) were dissolved in distilled water at 100mM and 25mM, respectively. Hemin was dissolved in 0.1M NaOH at 50mM. Human bioactive hepcidin was dissolved in water at 0.5mM, added to undifferentiated Caco-2 culture at 0.5uM, and incubated for 18–24 hr.

## Western blot and antibodies

Whole cell lysates (WCLs) were prepared from sub-confluent or confluent cells grown in 60 mm culture plates (Greiner, 628160). In some experiments, cells were treated with FAC, DFO, or hemin for 18–24 hr before harvest. Cells were washed with phosphate buffered saline (PBS: 137mM NaCl, 27mM KCl, 15mM $KH_2PO_4$, 81mM $Na_2HPO_4$) and lysed in 200-400ul of PBS-based RIPA buffer (1X PBS, 1% NP40, 0.5% sodium deoxycholate, 0.1% SDS) containing 1X protease inhibitor cocktail (Calbiochem/Millipore). In Fig 2, we also prepared HEK293 WCL in RIPA buffer sonicated for 10 sec. three times in ice-water (550 Sonic Dismembrator, Fisher) or WCL in transmembrane extraction reagent (Five Photon Biochemicals). We did not centrifuge WCLs to remove DNA or debris. Protein concentrations in WCLs were measured with 1X Bio-Rad protein assay dye reagent. 15–30 ug of WCLs were mixed with 2X SDS-PAGE

(polyacrylamide gel electrophoresis) sample buffer containing 62.5mM Tris-HCl (pH 6.8), 25% Glycerol, 2% SDS, 0.01% bromophenol blue, and 5% β-mercaptoethanol. The volume of 2X SDS-PAGE sample buffer was equal or more than varied WCL volumes for an equal amount of total proteins. For preparation of heated and non-heated samples, one group was heated at 95°C for 5min in a dry bath incubator (11-718-2, Fisher) containing water in a heating block, while another group was left at room temperature. After briefly spinning down sample tubes, WCLs were loaded on 10% SDS-PAGE (10% acrylamide, 0.27% bisacrylamide, 375mM Tris-HCl pH 8.8, and 0.1% SDS) mini-gel (4x2.9 inches). The stacking gel was 5% acrylamide containing 0.13% bisacrylamide, 125mM Tris-HCl pH 6.8, and 0.1% SDS. Protein size markers (Bio-Rad Precision Plus Protein Standards) was loaded without heating. The electrophoresis was run at 12.5mA per mini-gel for approximately 1.5hr in the running buffer containing 25mM Tris, 192mM Glycine, and 0.1% SDS. Proteins separated on the gel were transferred to PVDF membrane (Thermo Scientific, 88518) in a mini trans-blot cell (Bio-Rad, 022711PM) at 300mA in a transfer buffer (25mM Tris pH 8.3, 192mM Glycine, and 20% methanol) at 4°C for 1.5–2hr. The PVDF membrane was blocked for 20-30min at room temperature in either 1% BSA or 5% skim milk dissolved in 0.1% Tween 20-containing Tris-buffered saline (TBS: 20mM Tris-HCl pH 7.6 and 137mM NaCl) and incubated with a primary antibody on a rocker platform (Speci-Mix, Thermolyne) at 4°C overnight. The primary antibodies for western blots listed in Table 1 were diluted with either 5% skim milk in 0.1% Tween/TBS for anti-ferritin H (1:1,000), anti-TfR1 (1:20,000), anti-DMT1/SLC11A2 (1:1,000), anti-GAPDH (1:5,000), anti-Flag (1:1,500), or 1% BSA in 0.1% Tween/TBS for anti-Fpn1/SLC40A1 (1:1,000~1,500). The PVDF membranes were washed with approximately 50 ml of 0.1% Tween/TBS for 10-20min, three times at room temperature, and incubated with a secondary antibody as listed in Table 1 (5,000–8,000-fold diluted with the same dilution solution as a primary antibody, either 5% skim milk or 1% BSA in 0.1% Tween/TBS) for 1.5hr at room temperature. The PVDF membranes were washed with 0.1% Tween/TBS three times, incubated with ECL reagents (Table 1) at room temperature, and exposed to X-ray films (ProSignal Blotting Film, Genesee Scientific 30-507L or 30-810L). For successive incubation with another primary antibody, the membrane was soaked at room temperature for 1 hr in the stripping solution (pH 2.2 with HCl) containing 1.5% glycine, 0.1% SDS, and 1% Tween 20 (from Abcam's western blot membrane stripping for restaining protocol). After washing membrane in 0.1% T/TBS for 15min three times, the procedure was repeated from blocking the membrane with 1% BSA or 5% skim milk in 0.1% T/TBS.

## Isolation of cytoplasmic and nuclear fractions

Cells were washed with PBS, resuspended in 200-500ul hypotonic buffer (20mM Tris-HCl, pH 7.4, 10mM NaCl, 3mM MgCl$_2$) at 4°C for 15min, then mixed with 1/20 volume of 10% NP40, vortexed for 10 sec, and centrifuged at 12,000 rpm for 30 sec. The supernatant was recovered (cytoplasmic fraction), and the pellet was resuspended in 30-50ul of extraction buffer containing 100mM Tris, pH 7.4, 2mM Na$_3$VO$_4$, 100mM NaCl, 1% Triton X-100, 1mM EDTA, 10% glycerol, 1mM EGTA, 0.1% SDS, 1mM NaF, 0.5% deoxycholate, 20mM Na$_4$P$_2$O$_7$, and 1X proteinase inhibitor cocktail. The pellet suspension was vortexed for 10 sec, incubated on ice for more than 30min, vortexed for 30 sec, and centrifuged at 12,000 rpm for 10min at 4°C. The supernatant (nuclear fraction) was recovered.

## Plasmids, siRNA, and transfection

pCMVSport6Fpn1 (MGC human SLC40A1 sequence-verified cDNA, clone ID: 5213437, MHS6278-202801506) was purchased from GE Dharmacon. pBluescriptR-DMT1 (SLC11A2 MGC human clone ID: 5300266, MHS1010-98075408) was purchased from Open Biosystems.

The 2.5kb DMT1 cDNA was isolated after digestion with NcoI and HindIII followed by filling-in with DNA pol. Klenow fragment, and cloned into SmaI site of pCMV and pCMVFlag vectors. These plasmid DNAs, verified by DNA sequencing (Eurofins), were transiently transfected into HEK293 cells by electroporation (X-Cell, Bio-Rad) as described previously [36].

Human Fpn1 siRNA was purchased from Sigma-Aldrich (GAUGGAACUUGGUAUCCAU[dT][dT], and AUGGAUACCAAGUUCCAUC[dT][dT]) and 5ul of 20uM solution was transfected into Caco-2 cells (grown in 2ml media/35 mm plate) with 200ul of Opti-MEM containing 6ul of Lipofectamine RNAiMax (Invitrogen) for 18hr, followed by treatment with 250 and 500uM FAC for additional 8hr in the presence of siRNA prior to harvest. Knockdown of Fpn1 in Caco-2 cells was first verified by qPCR as described previously [37] using three sets of human Fpn1 primers.

Set1 forward: 5'-CTACTTGGGGAGATCGGATGT-3'

Set1 reverse: 5'-CTGGGCCACTTTAAGTCTAGC-3'

Set2 forward: 5'-TGGATGGGTTCTCACTTCCTG-3'

Set2 reverse: 5'-GTCAATCCTTCGTATTGTGGCAT-3'

Set3 QuantiTect Primer Assays for Hs_SLC40A1 (QT00094843, Qiagen)

## Results

### Routine practice of heating samples in Fpn1 (SLC40A1) western blot

According to the citation history for Fpn1 antibody in CiteAb, the rabbit polyclonal anti-Fpn1 antibody NBP1-21502 (Novus Biologicals) has been most successfully used in research article publications (https://www.citeab.com/). After preliminary testing of optimum dilutions of this primary antibody with either 5% skim milk or 1% BSA in Tween/TBS buffer, we decided to use 1,000–1,500-fold dilution in 1% BSA in Tween/TBS. We prepared WCL samples from several culture cell lines, heated at 95˚C for 5min, loaded on a 10% acrylamide SDS-PAGE gel, and performed western blotting. The blot gave rise to multiple bands including the putative Fpn1 band* (63 kDa) between 50 and 75 kDa size markers (Fig 1A).

The expression of Fpn1 can be increased by high iron levels through dissociation of IRE-binding protein 1 and 2 (IRP1 and IRP2) from the iron-responsive element (IRE) in 5'-UTR of Fpn1 mRNA [22]. Additionally, another Fpn1 mRNA (termed FPN1B) transcribed from an alternative promoter lacks the IRE but has the same open reading frame, therefore FPN1B expression is not responsive to cellular iron status [38]. To characterize the putative Fpn1 band further, we treated SW480 cells with ferric ammonium citrate (FAC) or cisplatin that directly inhibits IRP2 binding to IRE [39] and performed western blotting. The putative 63 kDa faint band was not increased by FAC but slightly increased by cisplatin treatment in SW480 cells (Fig 1B). In contrast, ferritin H was significantly increased by FAC and cisplatin because ferritin H mRNA also contains a functional 5'-IRE [40, 41], suggesting that FAC and cisplatin treatments are working to drive 5'-IRE-dependent translational upregulation. Collectively, we demonstrate that the most popular anti-Fpn1 antibody gave rise to the band around 63 kDa but it did not show clear regulation through the IRP-IRE system in SW480 cells. Thus, without taking a positive control for western blotting, the identity of the 63 kDa band in Fig 1 remained undetermined.

### No solubilization problem but heating samples may compromise Fpn1 western blot

To validate the putative Fpn1 band and anti-Fpn1 antibody further, we made WCL from HEK293 cells transiently transfected with a human Fpn1 expression plasmid and performed

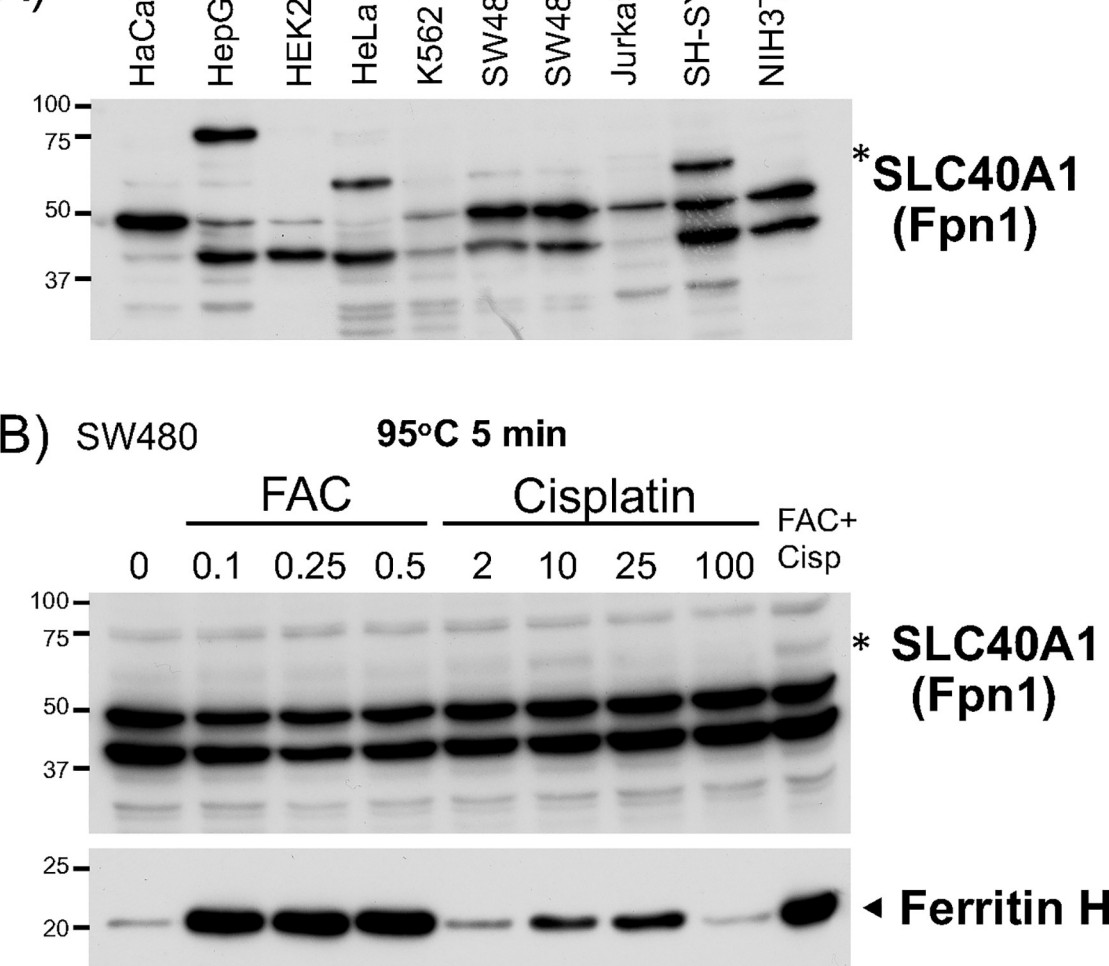

**Fig 1. Heated samples for Fpn1 (SLC40A1) western blotting. A)** 25ug of whole cell lysates (WCLs) prepared from indicated cells (*two different preparations of SW480 WCLs) were subjected to Fpn1 western blotting as detailed in the Materials and Methods. This experiment used sample loading buffer containing 5% β-mercaptoethanol, heated at 95˚C for 5 min, anti-Fpn1 antibody (Novus, rabbit polyclonal, 1,000-fold diluted with 1% BSA in T/TBS) at 4˚C overnight followed by anti-rabbit IgG-HRP conjugated (8,000-fold diluted with 1% BSA in T/TBS) at room temperature for 1.5hr. Unless otherwise noted, the rest of western blots in this study were performed in the same procedures with or without sample heating prior to gel loading. **B)** SW480 cells were treated with 0.1–0.5mM FAC (ferric ammonium citrate), 2-100ug/ml cisplatin, or FAC 0.25mM plus cisplatin 10ug/ml for 18hr. 20ug of WCLs were subjected to Western blotting with anti-Fpn1 antibody (top). The membrane was soaked in the stripping solution at room temperature for 1hr, washed with T/TBS, and incubated with anti-ferritin H mouse monoclonal antibody (sc-135667, Santa Cruz Biotechnology). In A) and B), the asterisks indicate the predicted position (63 kDa) of the endogenous Fpn1. The experiments were repeated three times and the representative western blots are shown.

western blotting in a routine sample preparation and heating protocol (see Materials and Methods). The overexpressed Fpn1 (untagged) gave rise to one specific band that was stuck on the top of the separation gel (Fig 2A).

This result indicates that the Fpn1 antibody (NBP1-21502, Novus Biologicals) can detect at least overexpressed human Fpn1 protein by western blotting, in which the Fpn1 protein appeared to be insolubilized or aggregated. To solve the potential solubilization problem first, we used three different cell lysis conditions; RIPA lysis buffer, RIPA plus sonication, and

transmembrane extraction reagent (Five Photon Biochemicals). To solve the aggregation problem, we compared heated samples at 95˚C to no heating (room temperature) of WCL from HEK293 transfected with pCMV-empty or -Fpn1 plasmid. As shown in Fig 2B, regardless of

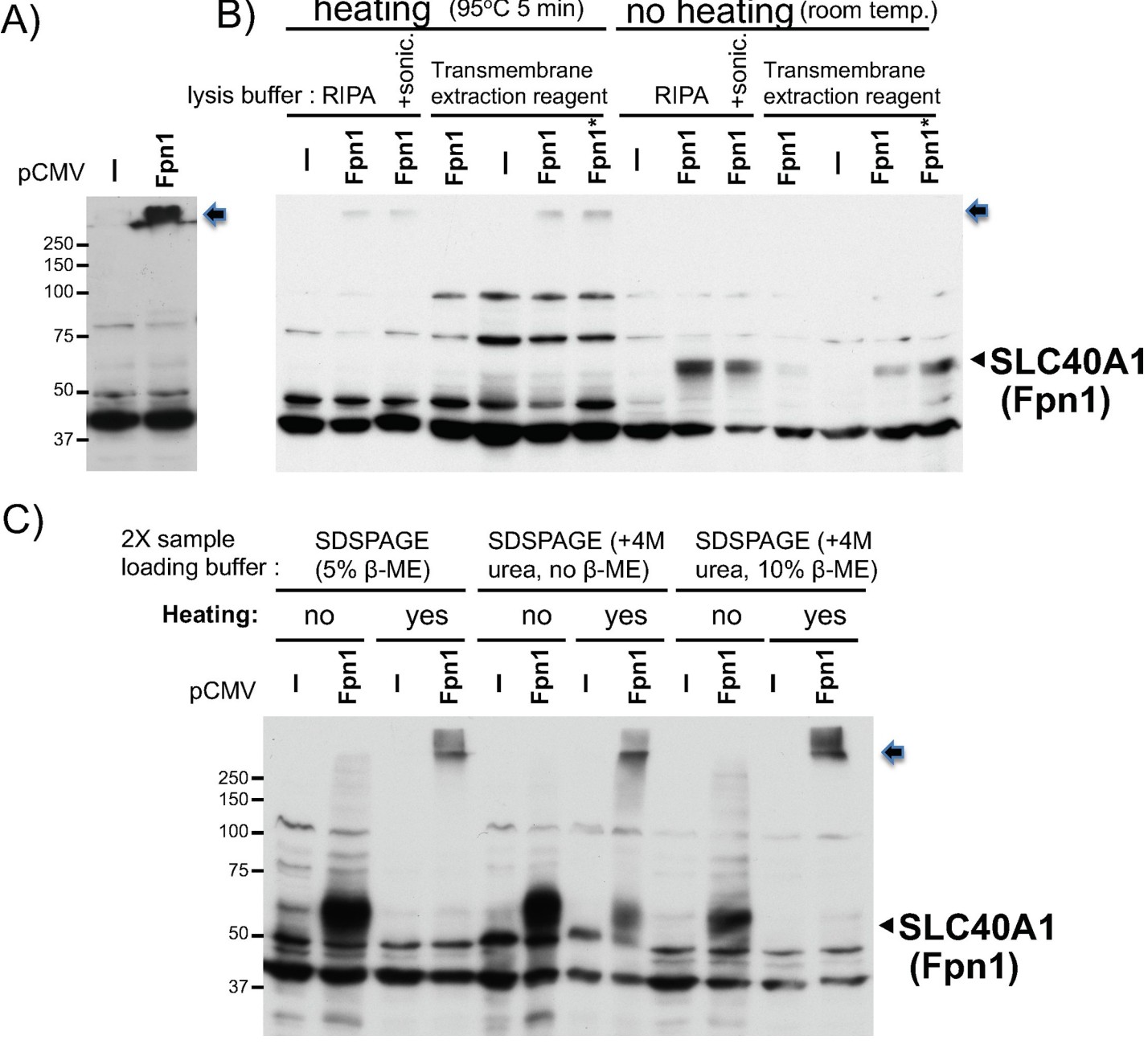

**Fig 2. Comparison in Fpn1 (SLC40A1) western blotting between heated and unheated protein samples from Fpn1 transfected cells. A)** WCL in RIPA buffer was prepared from HEK293 cells transfected with pCMV or pCVMFpn1. 25ug of the WCL mixed with 2X SDSPAGE sample loading buffer was heated at 95˚C for 5min and subjected to Fpn1 western blotting. The arrow indicates the Fpn1 protein stuck on the top of the separation gel (also in B and C). **B)** WCL in RIPA buffer, in RIPA plus sonication for 10 seconds three times in ice-water, or in transmembrane protein extraction reagent (Five Photon Biochemicals) was prepared from HEK293 cells transfected with pCMV or pCVMFpn1(6ug, *Fpn1: 12ug DNA). 15ug of the WCLs mixed with 2X SDSPAGE sample loading buffer were heated at 95˚C for 5min or not heated (at room temperature), and subjected to Fpn1 western blotting. The arrowhead indicates the transfected Fpn1 band. **C)** WCLs in RIPA buffer from HEK293 cells transfected with pCMV or pCVMFpn1 were mixed with 2X SDSPAGE sample loading buffer containing 5% β-mercaptoethanol (regular), 4M urea, or both 5% β-mercaptoethanol and 4M urea. One sample set was heated at 95˚C 5min, another set was not heated (at room temperature), and subjected to Fpn1 western blotting. Positions of molecular weight protein size marker are indicated on the left. The experiments were repeated three times and the representative western blots are shown.

three different lysis conditions, transfected Fpn1 was stuck on the top of the gel when they were heated at 95˚C prior to gel loading. In contrast, no heating samples gave rise to a specific Fpn1 band between 50 and 75kDa markers regardless of their different lysis conditions (Fig 2B). Furthermore, HEK293 WCL in RIPA mixed with three different 2X SDSPAGE sample loading buffers containing 5% β-mercaptoethanol (the regular sample loading buffer, see Materials and Methods), 4M urea, or both 5% β-mercaptoethanol and 4M urea did not improve the migration problem of overexpressed Fpn1 protein when they were heated (Fig 2C). By contrast, no heating of these WCL samples showed the right size of Fpn1 band regardless of the use of different loading buffers (Fig 2C). Taken together, we concluded that no heating of protein samples is the most critical factor for the right resolution and detection of transfected Fpn1 protein by western blotting with this anti-Fpn1 polyclonal antibody.

## Heating samples compromised DMT1 (SLC11A2) western blot

Another key SLC protein handling cellular iron traffic is DMT1 (SLC11A2). According to the citation data in CiteAb, there are more than 10 different anti-DMT1 antibodies used for publications. Among them, we used a rabbit monoclonal anti-DMT1 antibody (D3V8G, Cell Signaling Technology). To test whether heating protein samples also affects the resolution of DMT1 western blots, we prepared WCLs in RIPA or cytoplasmic and nuclear fractions from HEK293 cells transiently transfected with human DMT1 (untagged) or Flag-DMT1. Samples were mixed with 2x SDSPAGE sample loading buffer (containing 5% β-mercaptoethanol), heated at 95˚C for 5min or left at room temperature, and performed SDS-PAGE and western blotting with anti-DMT1 or Flag antibody. As observed in the Fpn1 western blots in Fig 2, heating protein samples significantly impaired the DMT1 protein bands (Fig 3A). In clear contrast, no heating of samples (left at room temperature until sample loading) showed good resolution of ~75kDa and additional larger and smaller bands detected with anti-Flag (Fig 3A, top) as well as anti-DMT1 antibodies (Fig 3A, bottom). DMT1 mRNA isoforms contain 3'-UTR IRE [42] that should be subject to mRNA destabilization when IRP1 and/or IRP2 dissociates from the IRE in iron overloaded conditions or by inactivation of IRP2 following cisplatin treatment [39]. Indeed, endogenous DMT1 proteins were detected by western blotting in unheated WCLs isolated from SW480 cells treated with FAC (0.1–0.5 mM) or cisplatin (2–100 ug/ml), in which FAC or cisplatin treatment slightly decreased DMT1 protein levels (Fig 3B). Collectively, no heating of protein samples prior to loading to SDS-PAGE gel is critical for detection of transfected and endogenous DMT1 proteins by western blotting.

## Effects of heating samples from 12 human cell lines on DMT1 and Fpn1 western blots

To assess the effect of heating samples on detection of endogenous DMT1 and Fpn1 proteins in various cell types by western blotting, we prepared WCLs from 12 human cell lines (lysed in RIPA), mixed with 2x SDSPAGE sample loading buffer, heated at 95˚C for 5min or not heated (left at room temperature), and performed SDSPAGE and Western blotting. Given the results of transfected Fpn1 and DMT1 in Figs 2 and 3, only unheated WCLs from HEK293 cells transfected with empty vector, DMT1, and Fpn1 were loaded together as positive controls of western blots. As shown in Fig 4A, endogenous DMT1 proteins were detected in most of these human cell lines; however, heating samples at 95˚C for 5min reproducibly impaired the major DMT1 band, some of which were stuck on the top of the gel (arrow on the right). Fpn1 western blots were more complicated because of multiple major bands in addition to no clear difference between heated and unheated samples (Fig 4B). This was an unexpected result because transiently transfected Fpn1 protein in HEK293 cells was detected only when the samples were

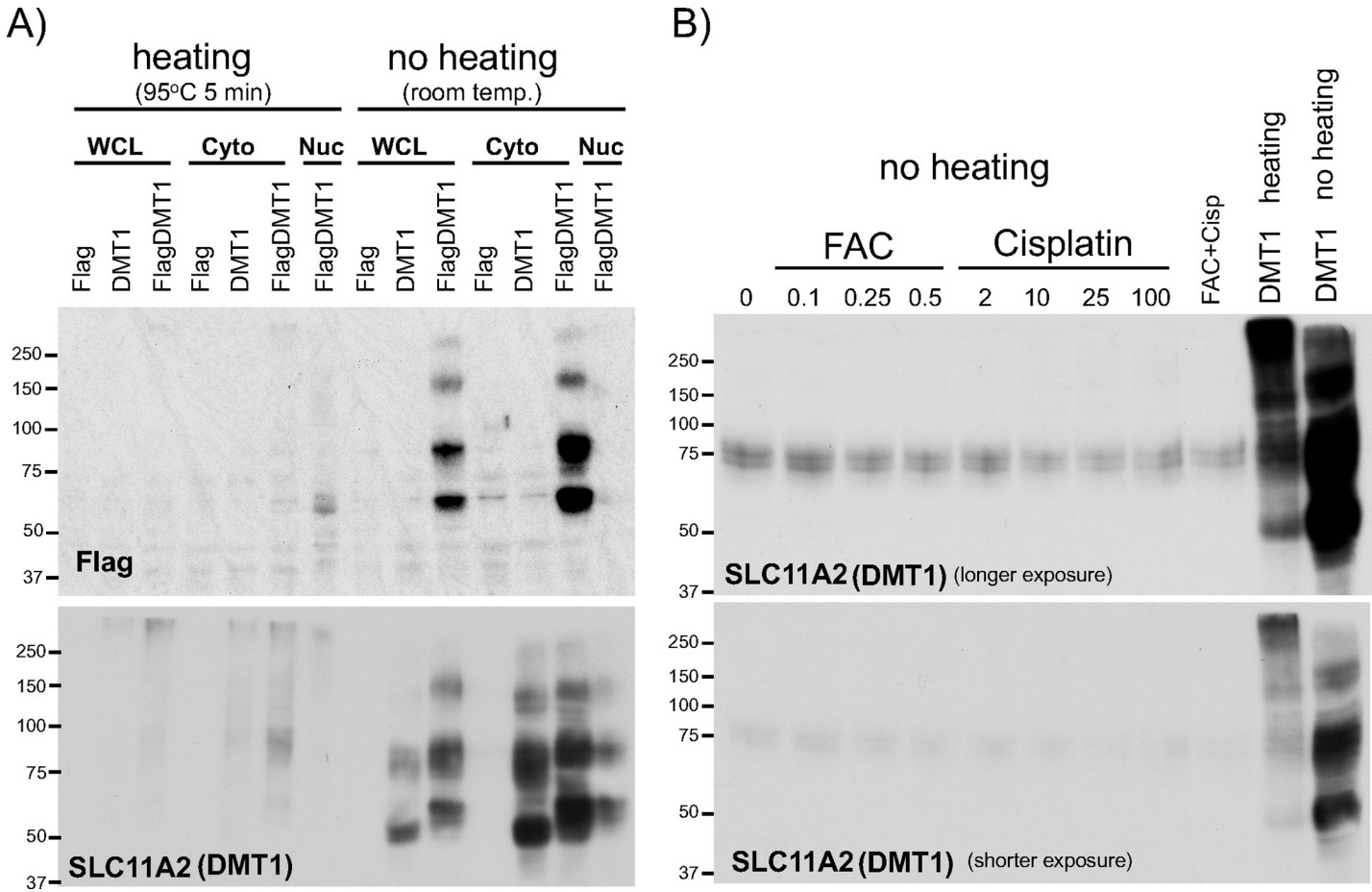

**Fig 3. Comparison in DMT1 (SLC11A2) western blotting between heated and unheated protein samples. A)** WCL in RIPA buffer, or cytoplasmic (Cyto) and nuclear (Nuc) fractions were prepared from HEK293 cells transfected with pCMVFlag, pCMVDMT1 or pCVMFlagDMT1. 30ug of the WCL, Cyto, and Nuc fractions mixed with 2X SDSPAGE sample loading buffer were heated at 95˚C for 5min or not heated (at room temperature), and subjected to western blotting with anti-Flag antibody (top) or anti-DMT1 antibody (bottom). **B)** WCLs in RIPA buffer were prepared from SW480 cells treated with 0.1–0.5mM FAC (ferric ammonium citrate), 2-100ug/ml cisplatin, or FAC 0.25mM plus cisplatin 10ug/ml for 18hr. 20ug of WCLs mixed with 2X SDSPAGE sample loading buffer without heating were subjected to Western blotting with anti-DMT1 antibody (top: longer exposure, bottom: shorter exposure). As a control, heated or unheated WCL from pCMVDMT1-transfected HEK293 cells was loaded. The experiments were repeated four times and the representative western blots are shown.

not heated prior to loading to SDSPAGE gel (Fig 2). The Fpn1 western blotting was working in Fig 4B because transiently transfected Fpn1 used as a positive control was detected in the range between 50 and 75kDa size markers (Fig 4B). The endogenous protein showing the same migration as transfected Fpn1 was also detected in several cell types such as Caco-2 cells (Fig 4B).

As our Fpn1 western blots gave rise to multiple non-specific bands, we further verified the endogenous band migrating together with transfected Fpn1. First, transfection of Fpn1 siRNA into undifferentiated Caco-2 cells reduced Fpn1 mRNA level to 35.3% (p = 0.007 in unpaired t-test, FAC 0) and consistently diminished the endogenous protein band co-migrating with transfected Fpn1 (Fig 5A). Second, as hepcidin binds Fpn1 and induces Fpn1 internalization and degradation [43] in Caco-2 cells [44], we treated Caco-2 cells (3 plates each) with bioactive hepcidin at 500 ng/ml for 24hr and performed Fpn1 western blotting, in which hepcidin-treated cells appeared to have decreased Fpn1 protein levels (Fig 5B). Collectively, we concluded that the band co-migrating with transfected Fpn1 is the endogenous Fpn1 protein.

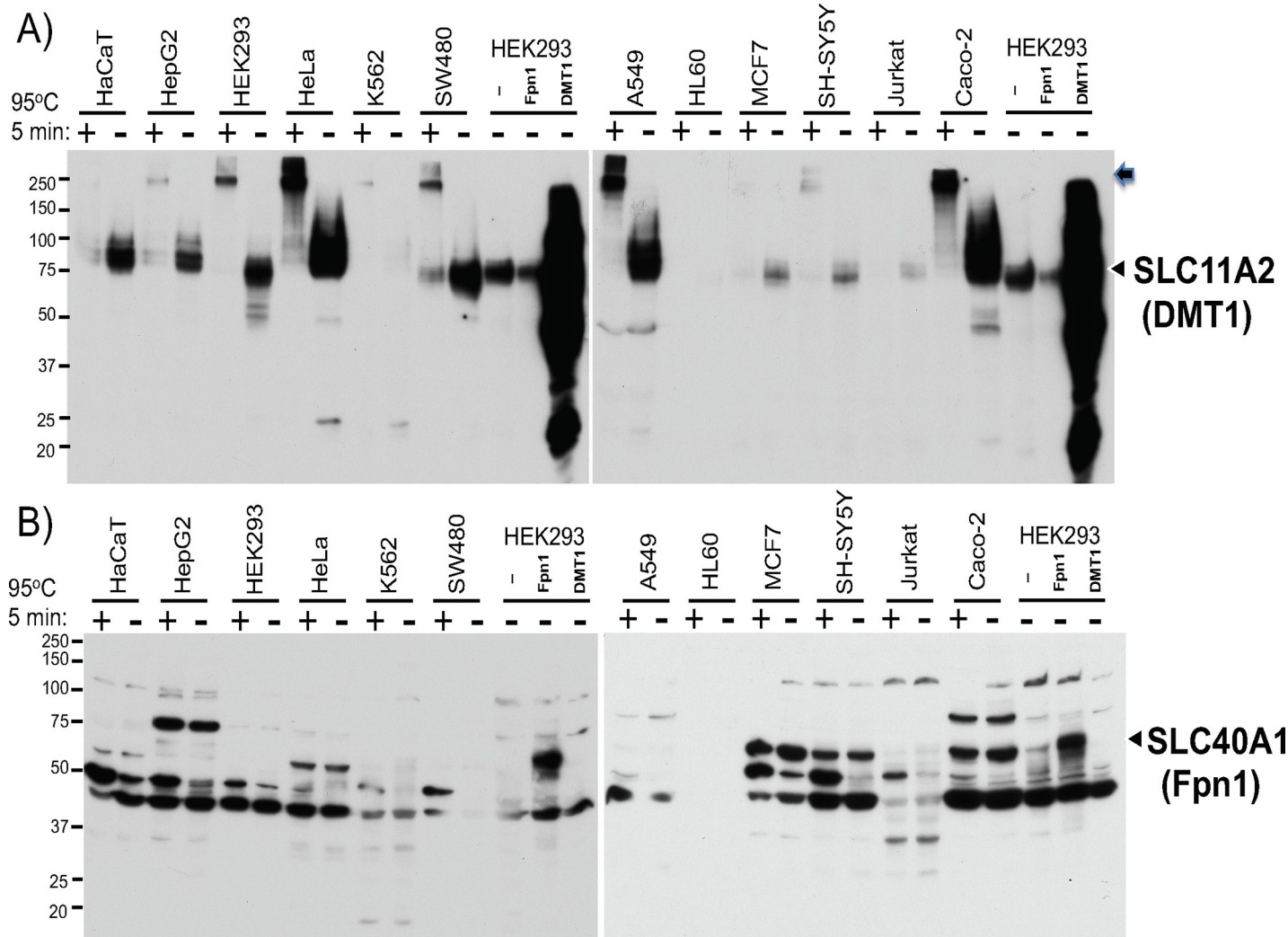

**Fig 4. Comparison in DMT1 and Fpn1 western blotting between heated and unheated protein samples from 12 human cell lines.** 25ug of WCLs in RIPA buffer from indicated 12 human cell lines were mixed with 2X SDSPAGE sample loading buffer, either heated at 95˚C for 5min or not heated (at room temperature), and subjected to **A)** DMT1 and **B)** Fpn1 western blotting. As a control, 10ug of non-heated WCL from HEK293 cells transfected with pCMV, pCMVDMT1, or pCMVFpn1 was loaded. The arrowheads indicate the positions of transfected DMT1 and Fpn1 bands. In A), the arrow indicates the DMT1 protein stuck on the top of the separation gel. The experiments were repeated four times and the representative western blots are shown.

## Effects of heating samples for single-transmembrane TfR1 and cytoplasmic ferritin western blots

We elucidated the effect of sample heating on western blots for two other key iron transport and storage proteins; single-transmembrane TfR1 and cytoplasmic ferritin. We prepared WCL in RIPA from Caco-2 cells treated with hemin, FAC, and an iron chelator deferoxamine (DFO) for 24hr. We compared heated samples with unheated samples for detection of each endogenous protein by western blotting. In Fpn1 western blotting, transiently transfected Fpn1 in HEK293 was reproducibly detected in unheated samples, whereas heated samples failed to show the right size of Fpn1 band between 50-75kDa protein size markers, rather showing very faint bands stuck on the top of the gel (arrow, Fig 6A right). FAC or hemin treatment in Caco-2 cells increased a band migrating similar to transfected Fpn1, in which the unheated group of samples showed slightly stronger bands compared to heated samples at 95˚C for 5min (Fig 6A).

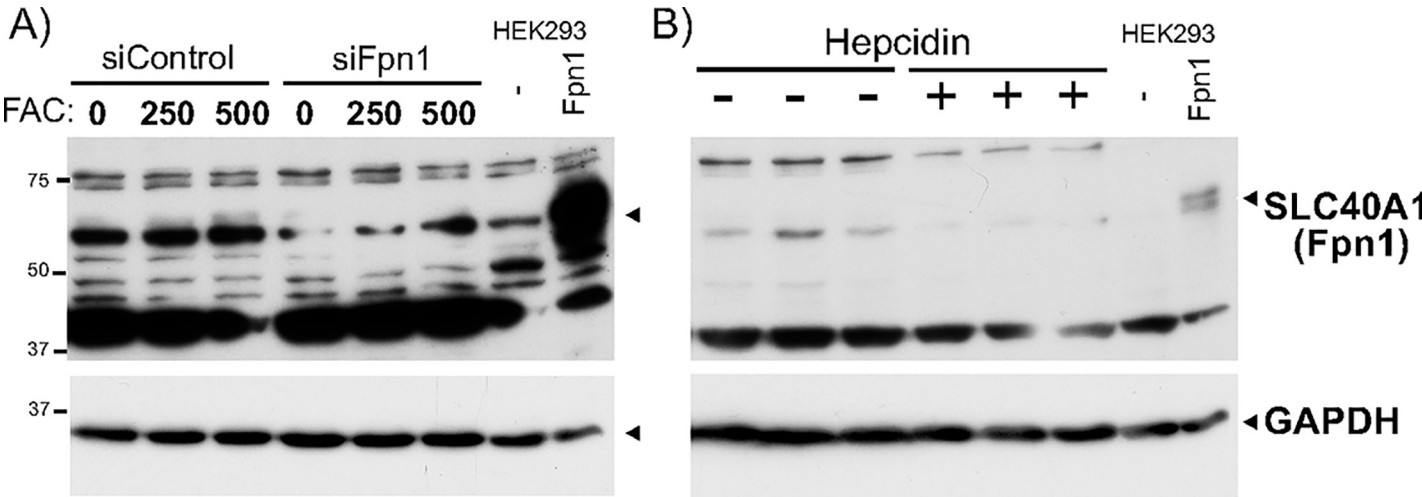

**Fig 5. Verification of the band specific to endogenous Fpn1 protein. A)** Caco-2 cells were transfected with Control- or Fpn1-siRNA using Lipofectamine RNAiMax for 18hr, followed by treatment with FAC at the final 250 and 500uM for additional 8hr in the presence of siRNA. Unheated Caco-2 WCLs in RIPA buffer along with unheated WCL from HEK293 cells transfected with pCMV or pCVMFpn1 were analyzed by western blotting for Fpn1 (top) followed by incubation with anti-GAPDH antibody. **B)** Caco-2 cells were untreated or treated with 500nM hepcidin (3 plates each) for 22h. 20ug of three-independent hepcidin (-) and hepcidin (+) WCLs in RIPA buffer were mixed with 2X SDSPAGE sample loading buffer, unheated, and subjected to Western blotting with the anti-Fpn1 antibody. Unheated WCL from HEK293 cells transfected with pCMV or pCVMFpn1 was loaded as a control. The membrane was incubated with the anti-GAPDH antibody to assess an equal amount of sample loading. The experiments were repeated five times and the representative western blots are shown.

Conversely, we anticipated that the iron chelator DFO would inhibit translation of Fpn1 mRNA via the 5'-UTR IRE; however, we repeatedly failed to observe it (Fig 5A, DFO compared to none). The endogenous DMT1 was consistently detected only when samples were not heated (Fig 6B). In contrast, the H subunit of ferritin (ferritin H) was detected as increased by hemin or FAC treatment only in heated samples (Fig 6C). As ferritin is composed of 24 subunits of ferritin H and L, heat denaturation of samples is necessary for ferritin western blot otherwise we often observed no ferritin bands or ferritin stuck on the top of the gel (S1 Fig). Interestingly, western blot for an 85 KDa homodimer of a plasma single-transmembrane protein TfR1 showed that unheated samples gave much stronger bands in an iron-responsive manner (iron-mediated downregulation) than samples heated at 95°C for 5min (Fig 6D). The unheated samples gave rise to an additional higher molecular weight band, probably a dimer of TfR1, in an iron-responsive manner (asterisk in Fig 6D). The cytoplasmic protein GAPDH as a protein loading control was detected equally in both heated and unheated samples (Fig 6E). Similar results were observed in HepG2 WCLs except for the endogenous Fpn1 band showing no clear differences between heated and unheated samples, both of which showed clear iron-induced Fpn1 expression following FAC treatment (S2 Fig).

## Discussion

Western blotting techniques were improved and seemingly have established, which may make us often omit in publications describing critical steps and tips for detection of protein of interest in western blotting. This is partly because a standard protocol for sample preparation has a routine protein denaturation step by heating at 95–100°C in a sample loading buffer containing SDS and a reducing agent such as β-mercaptoethanol or dithiothreitol. This heating step is understood important for antibodies to find epitopes efficiently as well as to detect a specific protein in multiple protein binding complexes or a subunit of multimeric proteins as exemplified by the detection of ferritin in this study (Fig 6C). However, detection of transmembrane

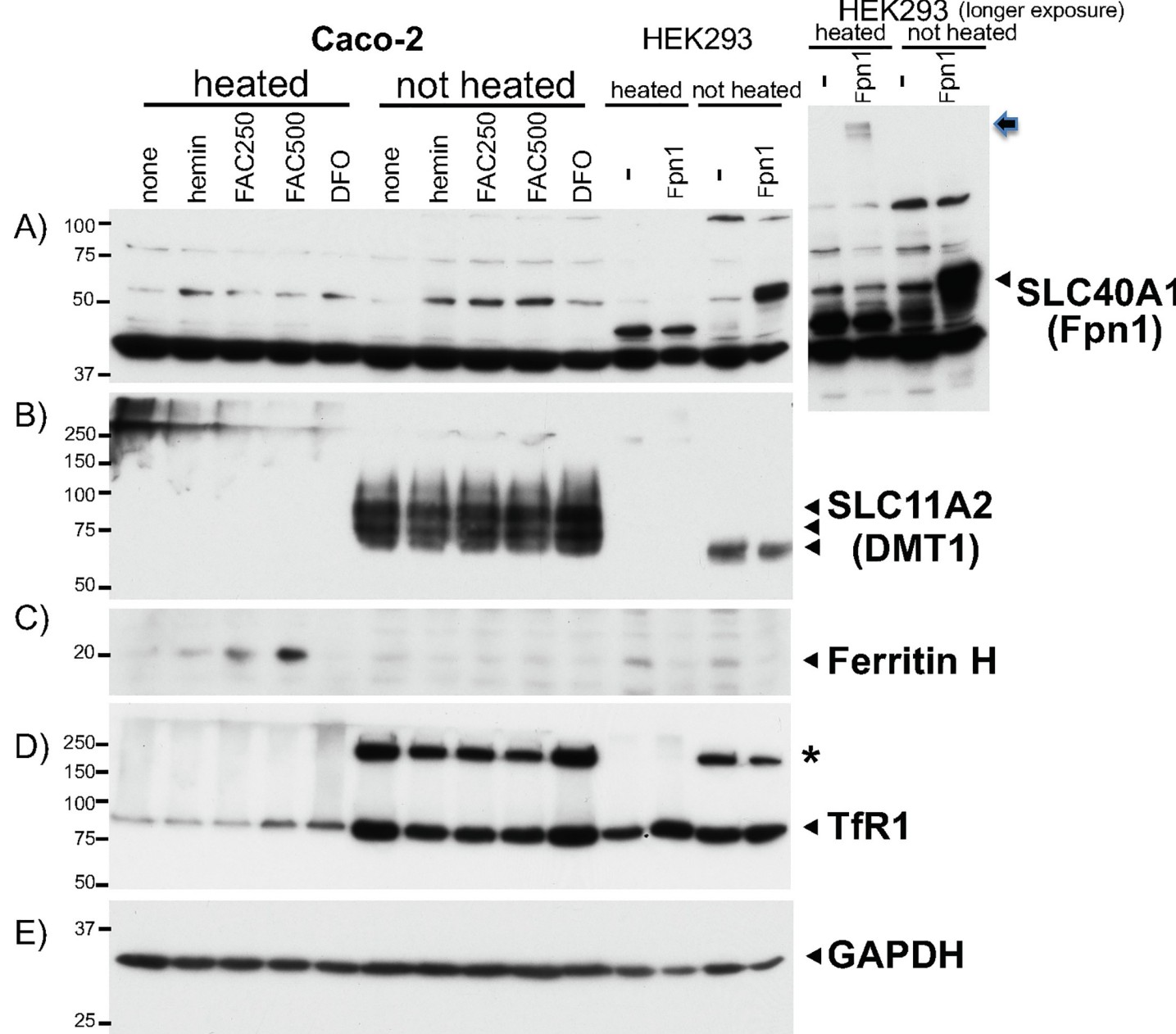

**Fig 6. Comparison in western blotting for ferritin and TfR1 proteins between heated and unheated samples from Caco-2 cells treated with iron.** Caco-2 cells were untreated or treated with 50uM hemin, 250 and 500uM FAC, or 25uM DFO for 24hr. 25ug of Caco-2 WCLs along with 10ug of WCL from HEK293 cells transfected with pCMV or pCVMFpn1 were heated at 95˚C for 5min or not heated (at room temperature). They were subjected to successive western blotting with **A)** anti-Fpn1 (SLC40A1), **B)** anti-DMT1 (SLC11A2), **C)** anti-ferritin H, **D)** anti-TfR1, and **E)** anti-GAPDH antibodies. The larger area of Fpn1 western blot for HEK293 cells transfected with pCMV or pCVMFpn1 is shown right next to the panel A. The arrow indicates the transfected Fpn1 protein stuck on the top of the separation gel. The asterisk in D) may be a TfR1 dimer only seen in unheated samples. The experiments were repeated three times and the representative western blots are shown. The same experiment repeated in HepG2 cells are shown in S2 Fig.

proteins by western blotting seems to be more challenging and complicated. There are more than 3,000 human genes encoding transmembrane proteins, in which ~ 1,300 receptors including seven-transmembrane GPCR (G protein coupled receptor) and single-pass transmembrane receptors, ~ 800 transporters such as SLC, ABC (ATP binding cassette) transporters, and channels, and ~ 500 enzymes including transferases, hydrolases, and oxidoreductases

[1]. We sometimes find researcher's troubleshooting discussions via internet medias about unsuccessful western blotting for detection of particular transmembrane proteins. Along with our initial failure to detect SLC family iron transporters in this study, we anticipate that routine heat denaturing protocols may not work or significantly loose band signals in western blotting to detect many more transmembrane and heat-sensitive proteins, perhaps due to heat-induced protein aggregations and/or conformational changes.

In this study, we observed that non-heated samples solely solved the migration and resolution problems of DMT1 and overexpressed Fpn1 western blots, whereas three different cell lysis conditions or three different 2X sample loading buffers did not improve the situation (Fig 2). Despite the fact that both DMT1 and Fpn1 are 12-transmembrane proteins [13, 15, 28, 29], heating samples at 95˚C for 5min ruined DMT1 western blot (Figs 3, 4A and 6B) whereas only partially impaired endogenous Fpn1 western blot (Figs 4B and 6A). However, heating samples completely impaired the migration of transfected Fpn1 (untagged) being stuck on the top of the separation gel (Fig 2A). We do not have good explanations for these different outcomes in sample heating between DMT1 and Fpn1 12-TMS proteins and between transfected and endogenous Fpn1 proteins. However, these results alert us to be aware of potentially different effects of sample heating on western blotting of any other SLC transporters and transmembrane proteins. It is therefore important to find the optimum condition of sample preparation for each transmembrane protein western blotting. This was also the case in TfR1 western blotting, in which non-heated samples tended to give rise to stronger and consistent iron status-dependent TfR1 band despite showing the additional TfR1 dimer band (Fig 6D, S2 Fig). In contrast, heating samples was essential for detection of cytoplasmic ferritin H subunit protein in western blotting because ferritin is assembled of 24 subunits of H and L proteins [11]. These results suggest that a single western blot loaded only either heated or non-heated protein samples cannot be used for successive reprobing with DMT1, Fpn1, TfR1, and ferritin antibodies. Another cytoplasmic protein GAPDH, often used as a protein loading control of western blotting, was not affected by heated or unheated protein samples. It should be noted that, although we observed no improvement for Fpn1 western blotting by switching from RIPA to RIPA plus sonication or commercially available membrane protein extraction lysis buffer (Fig 2A), the lysis buffer components together with mechanical destruction of membranes may also significantly improve western blotting for some other transmembrane proteins more than the impact of heating protein samples.

It may be a routine practice for laboratories and researchers primarily working on transmembrane proteins to avoid heat-denaturing protein samples prior to gel loading. Alternatively, they may have additional technical tips for particular transmembrane protein western blotting that we have not tested in this study. Indeed, no sample heating prior to gel loading for Fpn1 western blotting was noted in some earlier publications [22, 26, 28], while we observed relatively few attempts of DMT1 western blotting in earlier publications or no particular notes for western sample preparations except examples such as pre-incubation at 37˚C for 15min by Nam et al [45]. CiteAb records more than 50 citations for several applications with the rabbit polyclonal anti-Fpn1 antibody (NBP1-21502, Novus Biologicals) that we used in this study, in which more than 30 publications used this antibody for western blotting. Most of them did not clarify sample heating conditions or heated at 90–100˚C, except 75˚C for 10min [46] or no heating [39]. Even if protein samples might have been preheated in most of these publications, they might have observed the right size of the endogenous Fpn1 band as we observed in this study (Figs 4B and 6A). CiteAb also lists more than 10 publications for DMT1 western blotting using Proteintech, Abcam, or Novus Biologicals anti-DMT1 antibody, only some of which noted that they treated samples with gentle heating at 37˚C for 30min [47–49] or 50˚C for 15min [50]. Anti-DMT1 antibodies that have different epitopes from our anti-

DMT1 antibody (Cell Signaling Technology, near the N-terminus of human DMT1) may also affect the optimum conditions of protein sample preparation for better resolution of western blotting.

## Conclusion

In summary, only non-heated protein samples worked for western blotting with the anti-DMT1 (SLC11A2) monoclonal antibody having the epitope near the N-terminus. Non-heated protein samples gave rise to better resolution and/or stronger band signals in western blotting with anti-Fpn1 (SLC40A1) and anti-TfR1 polyclonal antibodies. Three different lysis buffers, three sample loading buffers, or sample sonication we tested did not improve the resolution of DMT1 and Fpn1 western blots. In contrast, only heated protein samples worked for ferritin western blotting with the anti-ferritin H monoclonal antibody, while sample heating did not affect western blotting for another cytoplasmic protein GAPDH frequently used for a protein loading control. Collectively, either heated or unheated samples most critically determined the quality and outcome of transmembrane protein western blotting we tested, suggesting that there should be many similar cases for western blotting of other transmembrane and heat-sensitive proteins.

## Supporting information

**S1 Fig. Unheated samples compromise ferritin western blotting.** WCLs in RIPA buffer were prepared from SW480 cells treated with 0.1–0.5mM FAC (ferric ammonium citrate), 10 or 25ug/ml cisplatin for 18hr. 20ug of WCLs mixed with 2X SDSPAGE sample loading buffer without heating and 25uM cisplatin WCL without or with heating at 95˚C, 5min were subjected to Western blotting with anti-ferritin H antibody, followed by incubation with anti-GAPDH antibody. The arrow indicates the ferritin protein stuck on the top of the separation gel.
(TIFF)

**S2 Fig. Comparison in western blotting for iron metabolism proteins between heated and unheated protein samples from HepG2 cells treated with iron.** HepG2 cells were untreated or treated with 100, 250uM FAC, or 25uM DFO for 24hr. Caco-2 cells were untreated or treated with 250uM FAC for 24hr. 30ug of HepG2, Caco-2 WCLs along with 10ug of WCL from HEK293 cells transfected with pCMV or pCVM Fpn1 were heated at 95˚C for 5min or not heated (at room temperature). They were subjected to successive western blotting with **A)** anti-Fpn1, **B)** anti-DMT1, **C)** anti-TfR1, **D)** anti-ferritin H, and **E)** anti-GAPDH antibodies. The arrow in A) indicates the transfected Fpn1 protein stuck on the top of the separation gel. The asterisk in C) may represent a TfR1 dimer.
(TIFF)

**S1 File. Original entire blots for figures.**
(PDF)

## Author Contributions

**Formal analysis:** Yoshiaki Tsuji.

**Funding acquisition:** Yoshiaki Tsuji.

**Investigation:** Yoshiaki Tsuji.

**Methodology:** Yoshiaki Tsuji.

**Project administration:** Yoshiaki Tsuji.

**Validation:** Yoshiaki Tsuji.

**Visualization:** Yoshiaki Tsuji.

**Writing – original draft:** Yoshiaki Tsuji.

**Writing – review & editing:** Yoshiaki Tsuji.

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
