## [Decision Letter · Decision Letter 0]

13 May 2020

PONE-D-20-06977

Transmembrane Protein Western Blotting: Impact of Sample Preparation on Detection of SLC11A2 (DMT1) and SLC40A1 (ferroportin)

PLOS ONE

Dear Dr. Tsuji,

Thank you for submitting your manuscript to PLOS ONE. After careful consideration, we feel that it has merit but does not fully meet PLOS ONE’s publication criteria as it currently stands. Therefore, we invite you to submit a revised version of the manuscript that addresses the points raised during the review process.

Please address all comments by reviewers, who are in general supportive of the publication of these results. There are two issues of concern, one related to repetitions/statistics, where I ask the authors for clarity above all. Likewise, there was some concern of the use of photoshop in some of the figures, please read carefully and adhere to PLOS One policy with respect to the images and state that you have done so in your reply. 

We would appreciate receiving your revised manuscript by Jun 07 2020 11:59PM. To enhance the reproducibility of your results, we recommend that if applicable you deposit your laboratory protocols in protocols.io, where a protocol can be assigned its own identifier (DOI) such that it can be cited independently in the future. For instructions see: http://journals.plos.org/plosone/s/submission-guidelines#loc-laboratory-protocols

We look forward to receiving your revised manuscript.

Kind regards,

Fanis Missirlis, Ph.D.

Academic Editor

PLOS ONE

Journal Requirements:

"Funding:

This work was supported in part by P30ES025128 from the National Institute of

Environmental Health Sciences to the Center for Human Health and the Environment

(CHHE)."

Reviewers' comments:

Reviewer's Responses to Questions

**Comments to the Author**

1. Is the manuscript technically sound, and do the data support the conclusions?

Reviewer #1: Partly

Reviewer #2: Yes

Reviewer #3: Yes

2. Has the statistical analysis been performed appropriately and rigorously? 

Reviewer #1: No

Reviewer #2: N/A

Reviewer #3: No

3. Have the authors made all data underlying the findings in their manuscript fully available?

Reviewer #1: Yes

Reviewer #2: Yes

Reviewer #3: Yes

4. Is the manuscript presented in an intelligible fashion and written in standard English?

Reviewer #1: Yes

Reviewer #2: Yes

Reviewer #3: Yes

5. Review Comments to the Author

Reviewer #1: PLOS ONE

Transmembrane Protein Western Blotting: Impact of Sample Preparation

Overview: A study investigating the effect of heating vs non-heating of protein samples prior to loading and its subsequent effect on detection of SLC11A2, SLC40A1 and TfR1.

Research groups have previously published SLC11A2 and SLC40A1 Western blots where the protein samples were heated prior to loading. Heating samples did not affect the bands in these studies 1,2.

The author describes in the discussion, research groups have also previously published western blots where the samples were not heated prior to loading. Therefore, I am uncertain about the novelty of this study.

Unfortunately, this was a very confusing paper to follow. The use of 12 cell lines to investigate the effect of heating may have been the reason for this confusion. My advice is to select a few cell lines which are involved in iron metabolism (i.e: Caco-2 and HepG2) and investigate what effect heating vs non-heating has on iron proteins from lysates obtained from these lines. Moreover, it would be useful to see if the effects observed are also present when other commercially available antibodies are used.

Further comments

Methods:

Please include a section about cell culture maintenance.

Please indicate how many days following transfection were the Caco-2 cells used?

Did the authors verify knockdown using qPCR?

Were Caco-2 cells undifferentiated/differentiated when lysed?

Results:

Please include figure captions.

1. Kondaiah P, Aslam MF, Mashurabad PC, Sharp PA, Pullakhandam R. Zinc induces iron uptake and DMT1 expression in Caco-2 cells via a PI3K/IRP2 dependent mechanism. Biochemical Journal. 2019;476(11):1573-1583.

2. Scheers NM, Almgren AB, Sandberg A-S. Proposing a Caco-2/HepG2 cell model for in vitro iron absorption studies. The Journal of nutritional biochemistry. 2014;25(7):710-715.

Reviewer #2: In the present work, Author Yoshiaki Tsuji investigated the impact of sample preparation for an optimal detection by western blotting of specific transmembrane proteins, SLC11A2 (DMT1) and SLC40A1 (ferroportin).

This work is based on initial observation of the Author concerning a mediocre functioning of SLC11A2 and SLC40A1 antibodies in his routine western blotting protocol. In the MS Author nicely and adequately explains, step by step, how specific modifications in sample preparation lead to significant improved western blot resolution for membrane proteins.

As far as technical issue for an optimal study of transmembrane protein is concerned, the MS deals about a subject of great interest and concerns proteins involved in iron metabolism, for which Author’s expertise is well established.

That said I have two minor points to address to the Author:

- Discussion, first page. The sentence beginning with “Given researcher’s troubleshooting discussions…” should be rephrased because it is too long and complex in this present form.

- I understand protocols were tested on different cell lines. Were lysates from primary cell cultures been also tested?

Reviewer #3: The manuscript addresses the question how different sample preparation has major affects on the outcome of Western blotting of several proteins related to iron metabolism. This is a worthy study as it can save many researchers a lot of pain, and improve study-outcomes.

Remarks

The authors relate in the abstract to DMT1 as an iron exporter. In most cases it though functions as an importer, when viewing the cytosol of a cell as “in” and the endosome as a part of internalized “out”. Later in the text DMT1 is termed a “transporter”, which does it more justice.

It would be much clearer to use Fpn and DMT1 throughout the paper, and not label the figures in one way and use in the text the SLC terms, sometimes in full and sometimes shortened e.g. SLC40.

In the text, a reference to fig. 2A is missing.

In Figure 3 B, the evidence for a decrease of DMT1 with increased FAC is weak and the decrease with increased Cisplatin is not convincing at all.

Can you please quantify these graphs and also indicate how often each experiment was repeated.

Indicate the number of times the experiments were repeated throughout the manuscript, best in the legends.

There seems to be a discrepancy between Fpn results in fig. 6a and 4b. Any explanation?

Figure legend 1 has a different format than the other legends, giving lot’s of info on blocking and washes. Could be omitted. Giving these details in materials and methods is enough.

6. PLOS authors have the option to publish the peer review history of their article (what does this mean?). If published, this will include your full peer review and any attached files.

Reviewer #1: No

Reviewer #2: No

Reviewer #3: No

---

## [Author Response · Author response to Decision Letter 0]

17 Jun 2020

Response to Reviewers

Manuscript ID#: PONE-D-20-06977

Manuscript Title: Transmembrane Protein Western Blotting: Impact of Sample Preparation on Detection of SLC11A2 (DMT1) and SLC40A1 (ferroportin)

Corresponding Author: Dr. Yoshiaki Tsuji 

Reviewer #1

Overview: A study investigating the effect of heating vs non-heating of protein samples prior to loading and its subsequent effect on detection of SLC11A2, SLC40A1 and TfR1. Research groups have previously published SLC11A2 and SLC40A1 Western blots where the protein samples were heated prior to loading. Heating samples did not affect the bands in these studies 1,2. The author describes in the discussion, research groups have also previously published western blots where the samples were not heated prior to loading. Therefore, I am uncertain about the novelty of this study. Unfortunately, this was a very confusing paper to follow. The use of 12 cell lines to investigate the effect of heating may have been the reason for this confusion. My advice is to select a few cell lines which are involved in iron metabolism (i.e: Caco-2 and HepG2) and investigate what effect heating vs non-heating has on iron proteins from lysates obtained from these lines. Moreover, it would be useful to see if the effects observed are also present when other commercially available antibodies are used. 

Response: Thank you for very constructive and helpful overview. 

First, we have never published SLC11A2 (DMT1) western blotting, retrospectively it was not successful because we routinely heated our samples and failed to detect convincing DMT1 bands. 

Second, yes this reviewer is right; we published SLC40A1 (Fpn) western blots using unheated samples in very low Fpn1-expressing cells (Cell Chem. Biol., 26:85-97, 2019). 

Third, as this reviewer pointed out, several papers showing DMT1 or Fpn1 western blots (in addition to the reviewer’s ref. 1 and 2) used heated samples (e.g. 100oC 10 min, 90oC 2 min, 75oC 10 min, 37oC 30 min), while most papers did not describe specific conditions for preparation of protein samples (discussed in this manuscript). I feel that some of published western blots for DMT1 (probably Fpn1 also) as well as some other SLC family members may not be accurate (particularly blots without indicating positions of protein size markers) or detected only partial quantities that did not aggregate even after heating, as seen in Figures 4 and 6. Please note that we have the following statement in the discussion (page 24) for PLOS ONE readers: “Indeed, no sample heating prior to gel loading for Fpn1 western blotting was noted in some earlier publications (22, 26, 28), while we observed relatively few attempts of DMT1 western blotting in earlier publications or no particular notes for western sample preparations except examples such as pre-incubation at 37oC for 15 min by Nam et al (44)”. 

Forth, all cell lines used in this study, including Caco-2 and HepG2, tightly regulate iron metabolism in response to environmental cues therefore have been used for investigation of the mechanisms of iron transport and storage. We hope that showing our results in all these cell lysates would help more researchers perform DMT1, Fpn1, or other transmembrane protein western blots with specific antibodies used in this study as well as their specific antibodies. 

We appreciate all of these reviewer’s important points.

Comment 1.1 Methods: Please include a section about cell culture maintenance.

Response: A new section of “Cell Culture and Maintenance” has been created in the Materials and Methods (page 6).

Comment 1.2 Please indicate how many days following transfection were the Caco-2 cells used?

Response: We have included the experimental timeline in the sub-section of Plasmids, siRNA, and Transfection in the Materials and Methods, page 12). 

Comment 1.3 Did the authors verify knockdown using qPCR?

Response: Yes, we verified Fpn1 knockdown in untreated Caco-2 cells in the experiment shown Fig. 5A. The qPCR results (35.3%, p=0.007: Fpn1 mRNA with transfection of siFpn1 compared to siControl) were included in the main text (page 18) without showing the graph in order to stay focus on technical tips in transmembrane western blotting. The qPCR methods and primer sets have been included accordingly in the Materials and methods (page 11). 

Comment 1.4 Were Caco-2 cells undifferentiated/differentiated when lysed?

Response: They were undifferentiated. In the revised manuscript, we have added a note that “undifferentiated Caco-2 cells were used in this study” in the “Cell Culture and Maintenance” section (page 6) along with in the main text (page 18). Thank you.

Comment 1.5 Results: Please include figure captions.

Response: Thank you, we have included figure captions in all figures.

1. Kondaiah P, Aslam MF, Mashurabad PC, Sharp PA, Pullakhandam R. Zinc induces iron uptake and DMT1 expression in Caco-2 cells via a PI3K/IRP2 dependent mechanism. Biochemical Journal. 2019;476(11):1573-1583.

2. Scheers NM, Almgren AB, Sandberg A-S. Proposing a Caco-2/HepG2 cell model for in vitro iron absorption studies. The Journal of nutritional biochemistry. 2014;25(7):710-715.

Reviewer #2

In the present work, Author Yoshiaki Tsuji investigated the impact of sample preparation for an optimal detection by western blotting of specific transmembrane proteins, SLC11A2 (DMT1) and SLC40A1 (ferroportin). This work is based on initial observation of the Author concerning a mediocre functioning of SLC11A2 and SLC40A1 antibodies in his routine western blotting protocol. In the MS Author nicely and adequately explains, step by step, how specific modifications in sample preparation lead to significant improved western blot resolution for membrane proteins. As far as technical issue for an optimal study of transmembrane protein is concerned, the MS deals about a subject of great interest and concerns proteins involved in iron metabolism, for which Author’s expertise is well established. That said I have two minor points to address to the Author:

Comment 2.1: Discussion, first page. The sentence beginning with “Given researcher’s troubleshooting discussions…” should be rephrased because it is too long and complex in this present form.

Response: Thank you, I totally agree. In the revised manuscript, it has been divided into two sentences and corrected to “We sometimes encounter researcher’s troubleshooting discussions via internet medias about unsuccessful western blotting for detection of particular transmembrane proteins. Along with our initial failure to detect SLC family iron transporters in this study, we anticipate that…..” (page 22) 

Comment 2.2: I understand protocols were tested on different cell lines. Were lysates from primary cell cultures been also tested?

Response: I have not tested primary culture cells. I understand this reviewer’s point that inclusion of primary culture cells would be helpful to more researchers having the transmembrane western problem. I expect that this study will be basically applicable to primary cultures. Readers will first test heating and lysis buffer conditions in their primary culture samples as demonstrated in this study, which hopefully could provide them with solutions or a better idea of further trouble shooting. 

Reviewer #3

The manuscript addresses the question how different sample preparation has major affects on the outcome of Western blotting of several proteins related to iron metabolism. This is a worthy study as it can save many researchers a lot of pain, and improve study-outcomes. 

Comment 3.1: The authors relate in the abstract to DMT1 as an iron exporter. In most cases it though functions as an importer, when viewing the cytosol of a cell as “in” and the endosome as a part of internalized “out”. Later in the text DMT1 is termed a “transporter”, which does it more justice.

Response: We totally agree with the important point. The abstract has been revised as follows: 

transmembrane iron transporter proteins; SLC11A2 (divalent metal transporter 1, DMT1), SLC40A1 (ferroportin 1, Fpn1), and transferrin receptor-1 (TfR1),

Comment 3.2: It would be much clearer to use Fpn and DMT1 throughout the paper, and not label the figures in one way and use in the text the SLC terms, sometimes in full and sometimes shortened e.g. SLC40.

Response: This is another important issue we realized after reading the manuscript carefully. In the revised manuscript, we primarily use Fpn1 and DMT1 rather than SLC40A1 and SLC11A2, respectively. All changes have been yellow-highlighted. Thank you for very helpful suggestions.

Comment 3.3: In the text, a reference to fig. 2A is missing.

Response: We have confirmed a reference to Fig. 2A in the text (page 13).

Comment 3.4: In Figure 3 B, the evidence for a decrease of DMT1 with increased FAC is weak and the decrease with increased Cisplatin is not convincing at all. Can you please quantify these graphs and also indicate how often each experiment was repeated. Indicate the number of times the experiments were repeated throughout the manuscript, best in the legends.

Response: We agree with the point of the small DMT1 expression changes in Fig. 3B (and also Fig. 6B). As 1) the trend of slightly decreased DMT1 expression in response to iron treatment is consistent in Fig. 3B and 6B, and 2) this manuscript primarily focuses on technical tips in transmembrane western blotting, we have just tempered the original statement in the text by adding “slightly” decreased DMT1 protein levels (Fig. 3B) at page 16. The experimental repeats have been indicated in each figure legend.

Comment 3.5: There seems to be a discrepancy between Fpn results in fig. 6a and 4b. Any explanation?

Response: Thank you for paying attention to this point. Although there might be a slight difference between these Fpn1 western blots in Caco-2 cells, the trend was similar; namely, heating samples did not compromise Fpn1 western blotting except for transfected (overexpressed) Fpn1.

Comment 3.6: Figure legend 1 has a different format than the other legends, giving lot’s of info on blocking and washes. Could be omitted. Giving these details in materials and methods is enough.

Response: We agree with simplifying the figure legend 1. As suggested, we have dropped redundant information including blocking and washing procedures.

---

## [Editor Report · Decision Letter 1]

18 Jun 2020

Transmembrane Protein Western Blotting: Impact of Sample Preparation on Detection of SLC11A2 (DMT1) and SLC40A1 (ferroportin)

PONE-D-20-06977R1

Dear Dr. Tsuji,

We’re pleased to inform you that your manuscript has been judged scientifically suitable for publication and will be formally accepted for publication once it meets all outstanding technical requirements.

Kind regards,

Fanis Missirlis, Ph.D.

Academic Editor

PLOS ONE
---

## [Editor Report · Acceptance letter]

23 Jun 2020

PONE-D-20-06977R1 

Transmembrane Protein Western Blotting: Impact of Sample Preparation on Detection of SLC11A2 (DMT1) and SLC40A1 (ferroportin) 

Dear Dr. Tsuji:

I'm pleased to inform you that your manuscript has been deemed suitable for publication in PLOS ONE. Congratulations! Your manuscript is now with our production department. 

Kind regards, 

on behalf of

Dr. Fanis Missirlis 

Academic Editor

PLOS ONE